# Evaluating the predictive power of combined gene expression dynamics from single cells on antibiotic survival

Razan N. Alnahhas,[1,2] Virgile Andreani,[1,2] Mary J. Dunlop[1,2]

**ABSTRACT** Heteroresistance and persistence are examples of mechanisms that can allow otherwise drug-susceptible bacteria to survive and resume growth after antibiotic exposure. These temporary forms of antibiotic tolerance can be caused by the upregulation of stress response genes or a decrease in cell growth rate. However, it is not clear how the expression of multiple genes contributes to tolerance phenotypes. Using fluorescent reporters for stress-related genes, we conducted real-time measurements of expression prior to, during, and after antibiotic exposure. We first identified relationships between growth rate and reporter levels based on auto- and cross-correlation analysis, revealing consistent patterns where changes in growth rate were anticorrelated with fluorescence following a delay. We then used pairs of stress gene reporters and time-lapse fluorescence microscopy to measure the growth rate and reporter levels in cells that survived or died following antibiotic exposure. Using these data, we asked whether combined information about reporter expression and growth rate could improve our ability to predict whether a cell would survive or die following antibiotic exposure. We developed a Bayesian inference model to predict how the combination of dual reporter expression levels and growth rate impacts ciprofloxacin survival in *Escherichia coli*. We found clear evidence of the impact of growth rate and *gadX* promoter activity on survival. Unexpectedly, our results also revealed examples where additional information from multiple genes decreased prediction accuracy, highlighting an important and underappreciated effect that can occur when integrating data from multiple simultaneous measurements.

**IMPORTANCE** Transient increases in bacterial antibiotic tolerance can result in treatment failure despite an infection initially presenting as susceptible, presenting a significant challenge in antibiotic therapy. This phenomenon can also provide a window of opportunity for bacteria to acquire permanent genetic resistance mutations. Although understanding the underlying mechanisms of these antibiotic tolerance phenotypes is crucial for developing effective approaches to treatment, current approaches for studying these transient phenotypes have limitations. Here, we use fluorescent reporters to monitor the expression of genes involved in stress response over time, aiming to link expression with antibiotic survival outcomes. Our results reveal a counterintuitive finding: monitoring multiple gene reporters does not necessarily improve our ability to predict antibiotic survival outcomes compared to single gene reporters. This result emphasizes the need for a deeper mechanistic understanding of the relationship between stress response gene expression and antibiotic tolerance.

**KEYWORDS** heteroresistance, single cell, antibiotic tolerance, Bayesian inference, cross-correlation, time-lapse microscopy

Address correspondence to Mary J. Dunlop, mjdunlop@bu.edu.

Razan N. Alnahhas and Virgile Andreani contributed equally to this article. The author order was determined by the time of first contribution on the project.

The authors declare no conflict of interest.

See the funding table on p. 18.

Tolerance to antibiotics poses a barrier to the treatment of bacterial infections and complicates the development of new antimicrobial agents (1). In "traditional"

genetic resistance, a microbe gains a genetic mutation or acquires DNA that allows it to grow in the presence of an antibiotic. By contrast, transient phenomena like heteroresistance and persistence can produce a temporary, non-heritable increase in antibiotic survival that occurs in a subset of cells within a population (2, 3). In this study, we focus on cases where fluctuations in the expression of stress response genes allow individual cells to temporarily tolerate short periods of antibiotic exposure. Heteroresistance, persistence, and other tolerance strategies are believed to have evolved as bet-hedging mechanisms, where subpopulations of cells enter a phenotypic state that can temporarily survive stressors (4–6). After the antibiotic has been removed and the population is re-established, the resulting progeny have the same minimum inhibitory concentration (MIC) to antibiotics as the original cells. In addition, the re-formed population contains a mixture of cells with baseline levels of susceptibility and cells that temporarily exhibit higher antibiotic tolerance. These phenomena can lead to incomplete cell killing during treatment, resulting in re-growth after antibiotic removal. In clinical contexts, this can result in infections that present as sensitive but recur after treatment (7). In addition, these antibiotic tolerance mechanisms can extend cell survival during antibiotic treatment, facilitating the acquisition of genetic mutations that confer full resistance (8, 9).

Previous studies have focused on determining the phenotypic states that underlie heteroresistance, persistence, and other transient forms of tolerance using methods such as gene knockouts (10–12), transcriptomics (13, 14), and single-cell resolution fluorescent reporter measurements (15–17). Transcriptomic approaches are well-suited for capturing the expression levels of many genes at once, which is important because stress response pathways are intertwined and can be regulated by the same sigma factors (18). For example, studies have shown the emergence of heterogeneous stress response states in bacterial populations post-antibiotic treatment using single-cell RNAseq (14). However, experimental protocols for transcriptomics require cell lysis; thus, they present a single time point "snapshot" view of the cell state, and it is not possible to follow the same cell before and after antibiotic treatment. By contrast, fluorescent reporters and time-lapse microscopy can report the activity of genetic pathways and cell growth rates in real time (19). Studies in this area have revealed that noise in gene expression and growth rate can be linked to antibiotic survival (20–22). Using fluorescent reporters for genes of interest allows for comparison of the phenotypic state of the cells that survive or die from antibiotic treatment in the period prior to the presence of antibiotics (16), providing information about the cell history that precedes the antibiotic tolerance phenotype.

The use of fluorescent reporters in previous studies has mainly focused on the analysis of single reporters. However, unobserved genes may also fluctuate in expression, affecting the phenotypic state of the cells and contributing to antibiotic survival. Capturing the expression of more than one gene at a time using multiple fluorescent reporters would provide more information, in principle increasing our understanding of how combined expression values influence cell outcomes following antibiotic exposure. In this work, we asked whether increasing the amount of data, by including simultaneous measurements of two fluorophores and growth rate, could improve our ability to predict whether a cell would survive or die following antibiotic treatment. This question is motivated by prior studies that have demonstrated correlations between expression in genes involved in stress response and antibiotic survival (3, 5, 16), as well as a widespread understanding that antibiotic efficacy is strongly related to single-cell growth rates (7, 22). Therefore, we developed a dual reporter system in which we combined fluorescent reporters for genes of interest from two stress response pathways. These reporters provide temporal data, allowing us to examine the relationship between multiple reporters and growth rate, and their impact on antibiotic tolerance at the single-cell level.

To assess how, and if, our understanding of phenotypic states associated with antibiotic tolerance is improved with this combined single-cell resolution expression data, we focused on reporters for two genes involved in stress response, *gadX* and *recA*.

The *gadX* gene encodes an acid stress response transcription factor, and *recA* encodes a DNA repair protein. We selected these genes due to reports in the literature relating their expression to antibiotic survival. Specifically, we used the antibiotic ciprofloxacin, a fluoroquinolone that inhibits DNA gyrase and leads to cell death due to double-stranded DNA breaks (23, 24). GadX expression has been shown to correlate with increased ciprofloxacin tolerance (16, 21), and prior reports have demonstrated an overlap in the acid stress and antibiotic resistance pathways (15, 25). RecA plays a role in the SOS response of *E. coli* and DNA repair (26, 27), and *recA* knockout studies have shown increased susceptibility to ciprofloxacin (12, 28). In addition, we showed previously that fluorescent reporters driven by the promoters for *gadX* and *recA* ($P_{gadX}$ and $P_{recA}$) fluctuate over time in single cells (16). We were therefore interested in understanding whether there were patterns of expression of these two otherwise unrelated stress response pathways that could work synergistically to increase the probability that a cell survives antibiotic treatment. More generally, we asked whether measuring multiple reporters could improve our ability to predict the probability of cell survival.

To test this, we used the promoters $P_{gadX}$, $P_{recA}$, and a constitutive promoter, $P_{const}$, to drive the expression of either cyan or yellow fluorescent protein (CFP or YFP), creating all pairwise combinations of the three promoters. We used these dual reporters in *E. coli* to measure single-cell gene expression dynamics before and during ciprofloxacin exposure. We then categorized cell outcomes, identifying which cells survived or died in the period following ciprofloxacin treatment. This analysis allowed us to look at temporal relationships between the expression of two reporters simultaneously in the same cell prior to antibiotic addition. We observed a clear interdependence between growth rate and fluorescence; however, different reporters had unique relationships, indicating this effect is not solely due to dilution associated with cell growth. Cross-correlations between fluorescent reporter signals confirmed that the promoters fluctuate independently from each other. Next, we examined the relationship between the activity of the three promoters and the growth rate on cell survival after antibiotic treatment. Through a series of calibration experiments informing a Bayesian inference model, we inferred underlying promoter activities, allowing us to conduct pairwise comparisons across all dual reporters. Using these data in a causal inference model, we determined the strengths of causal relationships between the promoter activities and growth rate with antibiotic survival. The model provided insight into how the added information about cell state from an additional reporter affects survival predictions. We found that increased $P_{gadX}$ expression always positively impacts ciprofloxacin survival, whereas growth rate always has a negative influence. Counterintuitively, when we considered data derived from multiple reporters, we encountered situations where the predicted nature of causal interactions varied, with certain cases showing clear examples where adding more data led to less accurate results. Our results pose an important question about the effect of unobserved factors in similar studies and motivate the need for parallel mechanistic studies to improve our understanding of tolerance.

## RESULTS

### Dual reporter construct design

We built strains of *E. coli* MG1655 containing dual reporter plasmids with all possible combinations of the promoters for *gadX* ($P_{gadX}$), *recA* ($P_{recA}$), and a constitutive promoter ($P_{const}$) driving the expression of either cyan fluorescent protein (CFP) or yellow fluorescent protein (YFP) (Fig. 1A; Table S1). In total, we constructed nine dual reporters, where this set includes distinct reporters, such as $P_{gadX}$-CFP/$P_{recA}$-YFP, and duplicate reporters, such as $P_{gadX}$-CFP/$P_{gadX}$-YFP. This design provides redundancy in our measurements, for example, allowing us to conduct experiments with both $P_{gadX}$-CFP/$P_{recA}$-YFP and the flipped fluorophore variant $P_{recA}$-CFP/$P_{gadX}$-YFP. Our dual reporter plasmids provide a complete pair-wise representation, allowing for detailed single-cell resolution measurements of gene expression over time.

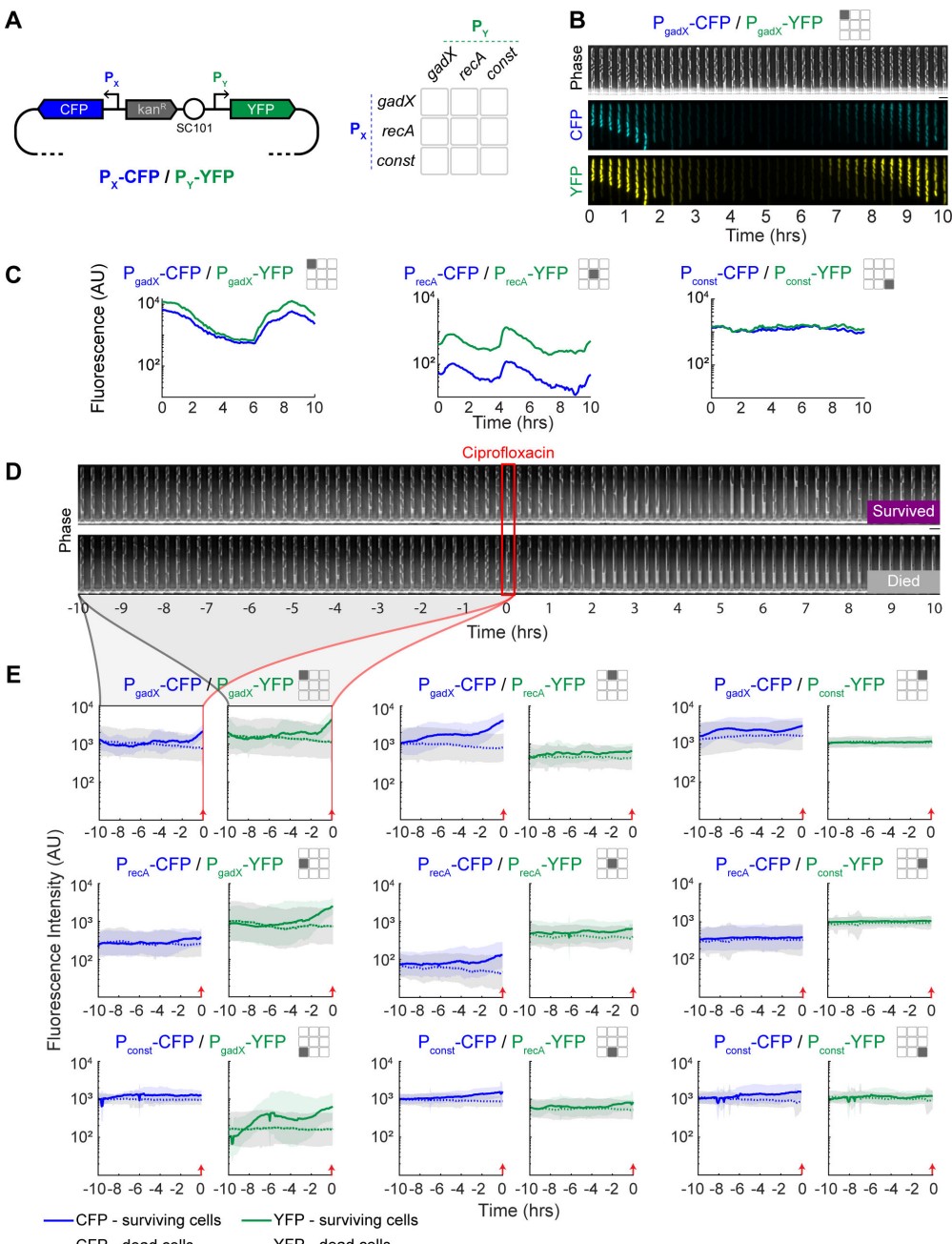

**FIG 1** Dual reporter levels over time in single cells and their relationship with ciprofloxacin survival. (A) Schematic of the dual reporter plasmid. Promoters for each fluorophore are oriented in opposite directions on the plasmid. The 3 × 3 grid depicts the pairs of promoters in the dual reporter plasmids, where the rows correspond to the promoter controlling CFP, and columns are YFP. This icon is used throughout the manuscript figures to specify which data are used. The diagonal values in this grid represent the duplicate reporters. (B) Kymograph of a representative cell with the $P_{gadX}$-CFP/$P_{gadX}$-YFP dual reporter in the mother machine device. Expression from the duplicate promoters is highly correlated. Scale bar is 4 µm. (C) Representative single-cell examples of fluorescence versus time data for each duplicate reporter plasmid. The $P_{gadX}$-CFP/$P_{gadX}$-YFP plot is the same cell as panel B. These data are all from the time period prior to antibiotic addition. (D) Kymograph showing representative cells that survived or died after antibiotic treatment. Time zero and the red box show the time of ciprofloxacin addition (1 µg/mL for 10 min). Scale bar is 4 µm. (E) Average fluorescence versus time data for cells with each reporter plasmid, where results are grouped into populations of cells that survived or died after antibiotic treatment. Data are separated by CFP (blue) and YFP (green). The mean fluorescence of cells that survive ciprofloxacin is shown in solid lines, and the mean fluorescence of cells that die after treatment is shown in dotted lines. Shaded regions represent ±1 standard deviation for

Fig 1 (Continued)

surviving cells in blue or green and for dead cells in gray. Time zero is the time of ciprofloxacin addition, indicated with a red arrow. Data for each reporter strain come from at least 100 cells from three or more biological replicates (Table S3).

We grew cells containing the dual reporters in the mother machine microfluidic device (29). This device constrains cells to grow in narrow, single-file lines where the mother cell remains trapped at the dead end of the chamber for the duration of the experiment (Fig. 1B). The device provides a constant flow of fresh media and the ability to add and remove antibiotics. We first grew the cells overnight in the mother machine while imaging with phase contrast and fluorescence every 5 minutes to establish a history of gene expression and growth for each cell. Using automated image analysis (30, 31), we extracted single-cell fluorescence and growth rates from the mother cells. Among the three reporters, $P_{gadX}$ had the highest level of variation in expression, while $P_{recA}$ exhibited moderate to low variation, and the constitutive control $P_{const}$ had very low variation across time (Fig. 1C). In addition, we confirmed that in the case of duplicate reporters such as $P_{gadX}$-CFP/$P_{gadX}$-YFP, changes in the expression of one fluorophore over time were also seen in the other fluorophore (Fig. 1C).

## Antibiotic survival experiments

After the period of overnight growth, we switched to media containing 1 µg/mL ciprofloxacin. We exposed the cells to ciprofloxacin for 10 minutes before allowing the cells to recover for 18 hours (Movies S1 to S9). This concentration and exposure time were selected based on our previous study (16), to produce a ~20% survival rate. We analyzed the time-lapse images to determine which mother cells survived and which died in the window following antibiotic exposure (Fig. 1D). We define surviving cells as those that resume division within 18 hours following ciprofloxacin removal, while any cells that did not resume division were categorized as having died.

We first asked whether there were differences in gene expression prior to ciprofloxacin addition that predisposed cells toward survival. To address this, we plotted the mean CFP and YFP reporter levels from the two categories (survived and died) over time for each of the nine dual reporter strains (Fig. 1E). We found that surviving cells, on average, had higher $P_{gadX}$ reporter levels in the period preceding ciprofloxacin addition, with differences emerging ~5 hours prior, suggesting that elevated $P_{gadX}$ expression correlates with survival. These findings show that the dual reporter system is consistent with our previous single reporter studies (16). By contrast, the $P_{recA}$ reporter levels showed only modest differences between the survived and died categories. As expected, the $P_{const}$ reporter levels typically did not correlate with survival. Unexpectedly, we did observe some fluorophore-dependent results where whether a promoter was controlling CFP versus YFP led to slightly different results, with the three $P_{const}$-CFP strains showing a modest correlation with survival, while the three constructs with $P_{const}$-YFP consistently showed no relationship. These results showing cell averages for the dual reporters motivated our focus on understanding how single-cell level expression dynamics of combined reporters and growth relate to survival.

## Temporal relationships between expression and growth

Given this comprehensive data set, we next quantified the temporal ordering between gene expression and changes in growth rate in single cells prior to antibiotic addition. For example, an increase in growth rate is likely to lead to a subsequent decrease in fluorescence due to dilution effects. The time delay between a change in growth rate and a change in fluorescence can be used to distinguish cause and effect. In addition, the maximum correlation or anticorrelation between the signals provides an additional indication of the relationship. For example, in the case of $P_{gadX}$, we observed a strong inverse relationship between single-cell growth and fluorescence, with changes in growth preceding changes in fluorescence (Fig. 2A). By contrast, while changes in $P_{recA}$

and $P_{const}$ were anticorrelated with growth after a delay, this effect was less pronounced than with $P_{gadX}$. The single-cell growth rates were smoothed with a moving average filter with a window of 1 hour to correct for noise because these raw values can be subject to numerical errors due to imperfect segmentation, and the smoothed value is a more accurate representation of the true growth rate (Fig. S1).

To systematically quantify these temporal correlations across all single-cell data, we calculated cross-correlations between the fluorescence and smoothed growth rate signals. If changes in fluorescence and growth are positively correlated, then the cross-correlation will have a positive peak, while negatively correlated signals exhibit a dip (Fig. 2B). In addition, if changes in fluorescence follow changes in the growth rate, we expect to see a positive delay in the cross-correlation curve, whereas if they precede changes in growth rate, then the time delay will be negative. Based on related studies, we largely expected to see anticorrelations between growth and fluorescence, with growth changes preceding fluorescence changes (16, 22, 32, 33), resulting in a positive time delay in the cross-correlation between fluorescence and growth rate.

We calculated cross-correlations for the time-series data from all nine dual reporters with growth, generating a total of 18 cross-correlation curves (Fig. 2C). Collapsing these data to include all cross-correlation results with identical promoters, for example, $P_{gadX}$-CFP derived from $P_{gadX}$-CFP/$P_{gadX}$-YFP, $P_{gadX}$-CFP/$P_{recA}$-YFP, and $P_{gadX}$-CFP/$P_{const}$-YFP, allowed us to identify consistent trends (Fig. 2D). We found that $P_{gadX}$ has a negative correlation with growth (maximum anticorrelation of −0.6) with a positive time delay (~1 hour) consistent with a reduction in reporter level following periods of increased growth rate. By contrast, while $P_{recA}$ and $P_{const}$ do exhibit a delayed anticorrelation with growth rate, the correlation is more modest (maximum anticorrelations of −0.2 to −0.4). This difference suggests a relationship between $P_{gadX}$ and growth that is due to more than just dilution. For example, $P_{gadX}$ is regulated by both RpoD and RpoS (34), and RpoS increases in expression during periods of slow growth (35). By contrast, $P_{recA}$ and $P_{const}$ are regulated by RpoD only (36–39). We also observed some fluorophore-dependent effects. While the overall trends in the cross-correlation curves were consistent, data derived from the CFP signals were more pronounced than those derived from YFP. We also observed some small positive values of the cross-correlations at negative time delays, primarily with the $P_{gadX}$ reporter. This may be the result of repeated patterns of growth and expression dynamics, which are more pronounced with $P_{gadX}$ than for the other two reporters. Overall, these data show anticorrelations between growth rate and each fluorescent reporter. Since the growth rate is known to affect antibiotic survival, we needed to take this relationship into account when assessing causal interactions between reporter levels and survival.

We also calculated the autocorrelation for each fluorescent reporter from the duplicate reporter constructs and compared it to fluorescence cross-correlations from these constructs. For example, we compared the autocorrelation of CFP or YFP data from $P_{gadX}$-CFP/$P_{gadX}$-YFP to cross-correlation of CFP and YFP data from $P_{gadX}$-CFP/$P_{gadX}$-YFP. Results were highly consistent, with this analysis showing very similar curves for all methods of calculation (Fig. 2E). In general, we found that the $P_{gadX}$ signals tend to have a slightly wider autocorrelation than $P_{recA}$ and $P_{const}$, which suggests that fluctuations in $P_{gadX}$ expression are slower than those observed in $P_{recA}$ and $P_{const}$. Calculations of the full width at half maximum to quantify autocorrelation width across all reporters confirms this general trend (Fig. 2F). Growth rate autocorrelations were also highly consistent regardless of the reporter strain from which the data were derived (Fig. S2) and were consistent with dilution due to cell division. We also calculated cross-correlations between the three different reporter fluorescence signals to examine temporal relationships between promoters (Fig. 2G). Distinct reporters were positively correlated, albeit at much lower levels than duplicate reporters (maximum correlations of ~0.4 compared to 0.9–1.0). We also found no evidence of temporal ordering in the fluorescence signals, with all cross-correlations exhibiting peaks at zero time delay. This

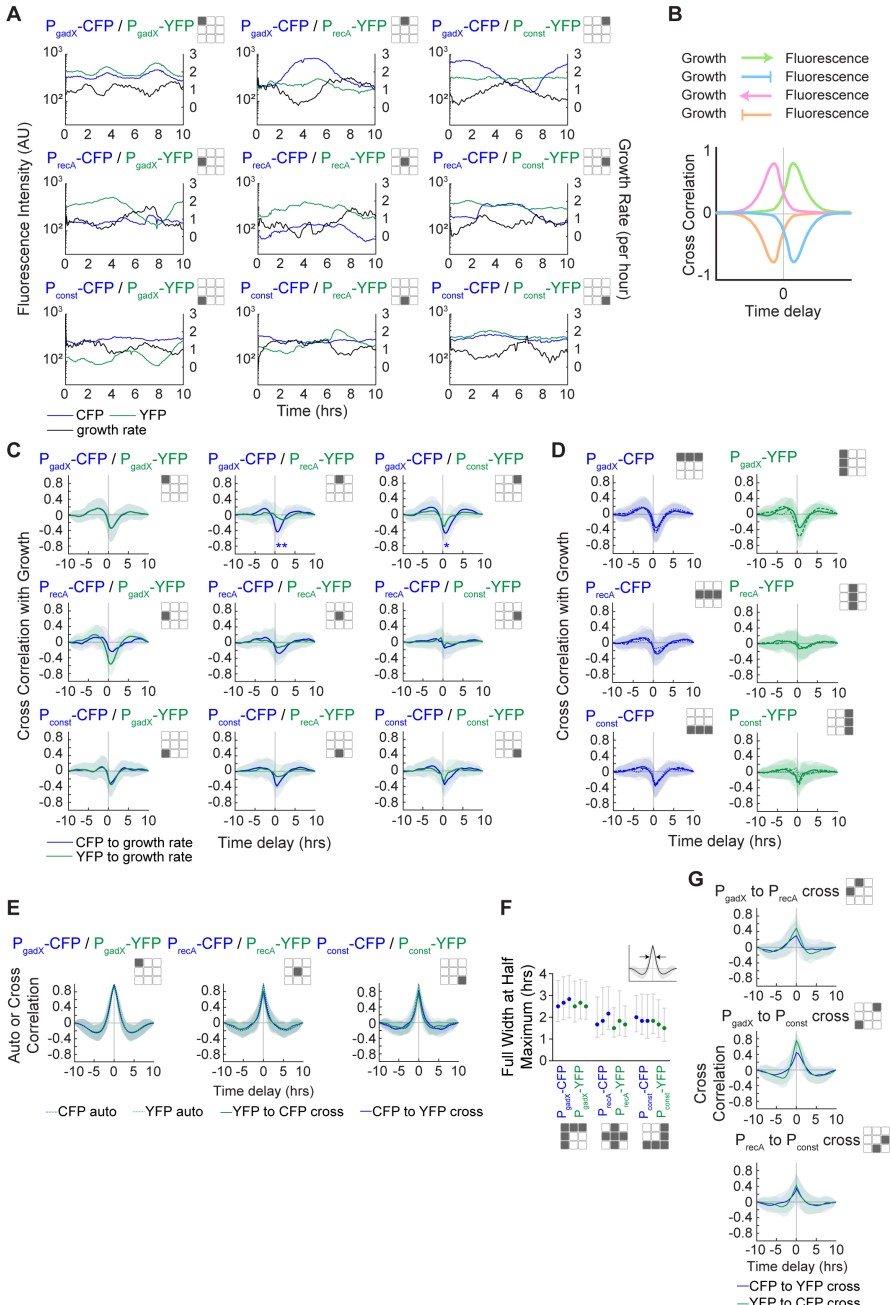

**FIG 2** Auto- and cross-correlations between fluorescent reporters and growth. (A) Representative fluorescence and growth rate data versus time from single cells for each dual reporter plasmid. These data are all prior to antibiotic addition. The growth rates in these and all analyses have been smoothed with a moving average filter with a window of 1 hour to correct for noise in growth rate calculations due to segmentation and numerical derivative calculations. (B) Schematic of the relationship between fluorescent reporters and growth rate for different directions and signs of the correlation. (C) Cross-correlation plots between fluorescence and growth rate for all dual reporter plasmids. For all plots, lines represent the average of all cells, and the shaded regions represent ±1 standard deviation. The blue lines in the grid represent the mean cross-correlation between the CFP reporter and growth, and the green lines represent the mean cross-correlation between the YFP reporter and growth. We performed a one-sample *t*-test to determine whether the value at the minimum of the cross-correlation significantly differed from zero: *$P < 0.05$, **$P < 0.01$. (D) Combined plots from panel C to plot all the P$_{gadX}$-CFP, P$_{recA}$-CFP, and P$_{const}$-CFP cross-correlations with growth (left) and all the P$_{gadX}$-YFP, P$_{recA}$-YFP, and P$_{const}$-YFP cross-correlations with growth (right). (E) Fluorescence auto- and cross-correlations of data from duplicate reporter plasmids with the same promoter driving CFP and YFP expression. Dashed lines represent autocorrelations, and solid lines represent cross-correlations. (F) Full width at half

**Fig 2 (Continued)**

maximum values for all reporters. Data are presented in groups corresponding to $P_{gadX}$, $P_{recA}$, and $P_{const}$. The inset shows a schematic of the full width at half maximum. Error bars show the maximum and minimum widths based on the standard deviation of the delay from the autocorrelation plots. (G) Fluorescence cross-correlations from distinct reporter plasmids with different promoters driving CFP and YFP expression. For all plots, lines show the mean of all cells, and the shaded regions represent ±1 standard deviation. All data in this figure are from conditions prior to antibiotic addition.

suggests the three reporters are under independent control, or that any upstream signals that affect them together operate on the same timescale.

## Ciprofloxacin survival as a function of promoter activity

Next, we focused our analysis on the influence of gene expression and growth rate on ciprofloxacin survival. Specifically, we were interested in asking whether expression data from $P_{gadX}$, $P_{recA}$, $P_{const}$; growth rate; or some combination of these signals would be sufficient to predict survival following antibiotic exposure. In addition to prior studies linking gene expression and antibiotic survival (3, 5, 16), there is widespread understanding that antibiotic efficacy is strongly related to single-cell growth rates (Fig. S3) (7, 22).

From the mother machine experiments, we captured the CFP and YFP reporter fluorescence values and growth rate for each cell and overlaid this information with data about whether the cell survived or died following ciprofloxacin treatment (Fig. 3A). For this analysis, we considered the CFP and YFP fluorescence values for each reporter at the time of antibiotic addition and the mean growth rate of the cell over the hour preceding antibiotic addition. For each dual reporter strain, we have data for CFP, YFP, and growth rate, generating a three-dimensional data set (Fig. 3A). To visualize this, we plotted three two-dimensional projections of this data set on each pair of variables in a triad of plots (Fig. 3B).

To address the question of whether combinations of measurements of expression and growth could predict antibiotic survival, we developed a model that considers the data from all the reporters at the same time in a unified framework. Because reporter data are derived from multiple constructs and different fluorophores, it was first necessary to perform a series of data transformations to estimate underlying promoter activities. For instance, $P_{gadX}$ data can be derived from $P_{gadX}$-CFP/$P_{gadX}$-YFP (in two possible ways), $P_{gadX}$-CFP/$P_{recA}$-YFP, and $P_{gadX}$-CFP/$P_{const}$-YFP, as well as from the equivalent constructs with the fluorophores swapped. An additional motivation for calculating promoter activities rather than using fluorescence measurements directly is that fluorescence measurements are a consequence but not a cause of changes in upstream pathways involved in antibiotic survival. Using fluorescence measurements as a direct observable of the current activity of these pathways would ignore that these values stem from earlier activity of the pathway from which they have had time to diverge. In addition, these transformations are necessary to remove the effect of growth rate on fluorescence through dilution. Thus, we set out to remove the influence of the biological and experimental processes occurring between the promoter activity and the fluorescence measurements, with the idea that the promoter activity responsible for the current fluorescence is a better representation of current pathway activity than the fluorescence itself.

We first accounted for the fact that we were required to use different imaging settings across the reporter strains and fluorophores due to differences in promoter strength (Table S2). Using calibration experiments, we determined that the influence of imaging settings on the measured fluorescence values was multiplicative (Fig. S4 and S5). We then modeled each CFP setting and each YFP setting as a multiplicative factor. The resulting values after compensating for the imaging settings only depend on the promoter and which reporter it controls; they could be biologically interpreted as representative of the number of fluorescent molecules per cell, independent of the way they were measured. However, this is still not a direct representation of the promoter activity, as the number of proteins in a cell is also affected by dilution and by the

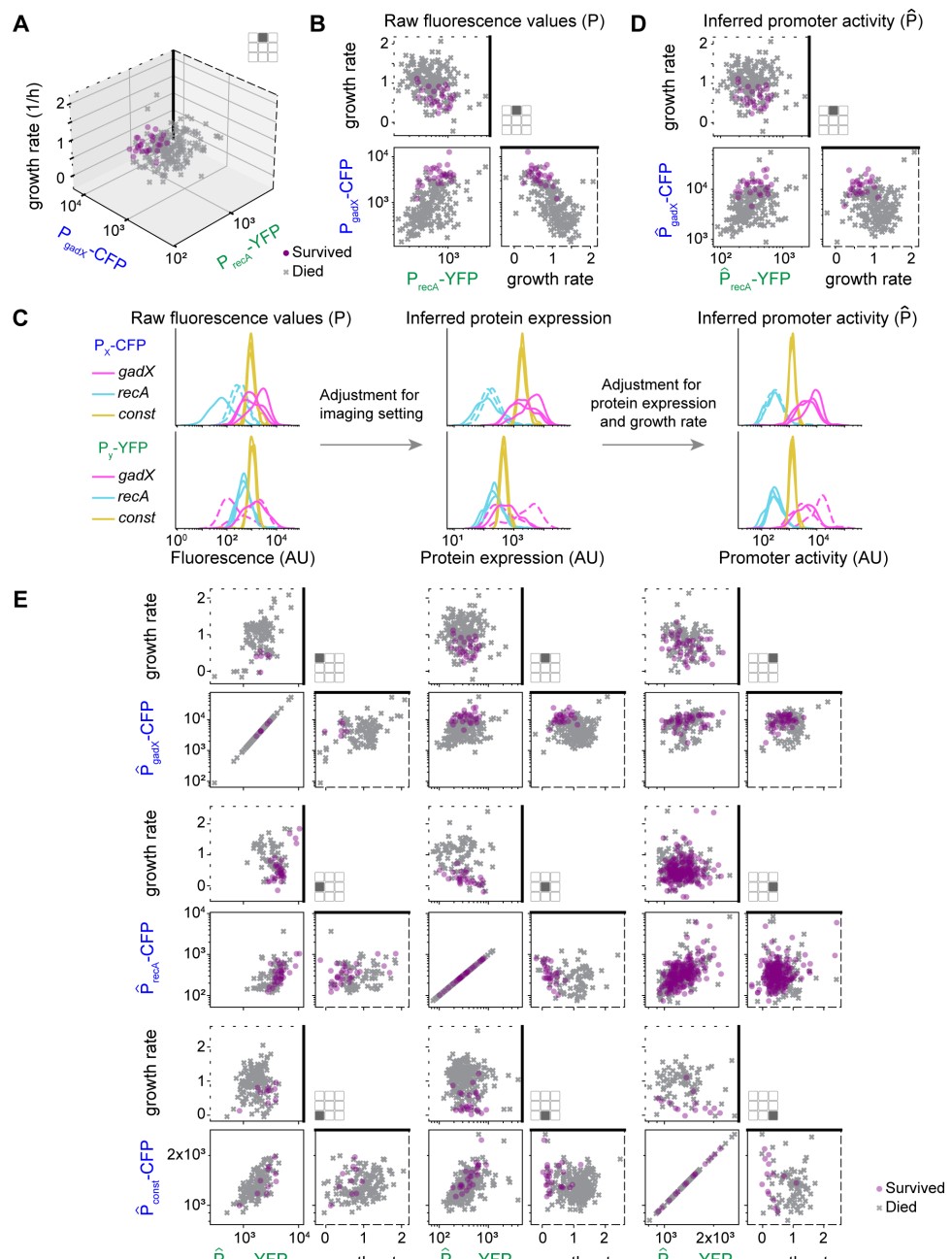

**FIG 3** Single-cell survival as a function of promoter activity and growth rate. (A) Three-dimensional plot of characteristics of cells that survived and died following antibiotic treatment. Data in this plot are from the strain $P_{gadX}$–CFP/$P_{recA}$-YFP. The two fluorescence values at the time of antibiotic exposure are plotted on the two horizontal axes, and the growth rate is plotted on the vertical axis. The growth rate is the average value from the last hour before antibiotic addition. Growth rate units 1/h; fluorescence units AU. (B) Three projections of the data from panel A are shown on three orthogonal planes in a triad of two-dimensional plots. The dashed outlines of the axes correspond to panel A, as a visual aid. (C) Data normalization process. Each of the nine constructs produces two fluorescence values per cell, CFP and YFP. For each construct and each color, the distribution of these values over cells is visualized with a kernel density estimation (smooth histogram). The left panel shows the raw fluorescence values, with CFP in the top row of plots and YFP on the bottom. The distributions are colored according to the promoter that drives this particular fluorescent reporter. The dashed lines correspond to constructs that have been measured with different imaging settings relative to the others (Table S2). The center panel shows the same distributions after adjustment for optical imaging settings. The right panel shows the distributions after adjustment for protein expression and growth rate. (D) This triad of scatter plots shows the inferred promoter activity ($\hat{P}$) corresponding to the raw fluorescence

Fig 3 (Continued)

values in panel B. Here, every cell was given values corresponding to the average of the posterior distributions. Growth rate units 1/h; promoter activity units AU. (E) Similar to panel D but with every dual reporter construct shown. Note that panel D appears on this panel in the top row, central column.

efficiency of transcription and translation. We took the same approach for these effects and determined coefficients to compensate for protein expression and dilution using a multiplicative model. After these two data transformations, we obtained values for each promoter activity (denoted $\widehat{P}$), independent of the imaging settings or dual reporter variant from which the data were sourced (Fig. 3C; Fig. S6). By performing these corrections, it is possible to directly compare data from each promoter, allowing us to compare results collected across varying reporters, growth rates, and imaging settings. As a check that this analysis is realistic, we verified that the activity of each promoter lined up when comparing the CFP and YFP channels and that all data from a single promoter were similar, regardless of the reporter strain from which they were derived (Fig. 3C, right panels). We also validated this analysis computationally on a simulated data set (Fig. S7 and S8). This series of transformations allowed us to directly compare data collected from different sources in a unified framework by generating inferred promoter activity values (Fig. 3D).

We converted data from all experiments into inferred promoter activities and compared results across the dual reporter strains (Fig. 3E). Note that for duplicate reporters with identical promoters (e.g., $\widehat{P}_{gadX}$-CFP/$\widehat{P}_{gadX}$-YFP), the distribution is diagonal, because there is a single value per promoter and per cell. Comparing the raw fluorescence measurements (Fig. S9) and the inferred promoter activity (Fig. 3E), we found that this data transformation significantly reduced the anticorrelation between fluorescence and growth rate, as expected given that we compensate for this with our model. In addition, we observed qualitative patterns of gene expression that correlate with ciprofloxacin survival, where the surviving cells tend to cluster in certain regions of the promoter activity and growth rate space. These patterns motivated us to ask whether there were specific expression and growth profiles that were associated with survival.

## Estimation of the direct causal effects of promoter activity and growth rate on survival

To test the hypothesis that the combined activity of the measured cellular pathways and growth at the time of antibiotic treatment can be used to predict survival, we used the inferred promoter activity values and growth rate in a logistic regression where the output is the probability of cell survival. The complete conceptual model can be represented as a directed acyclic graph (Fig. 4A), where the direction of the arrows represents causality. Note that although we use regular arrowheads in our depictions, both positive and negative effects can be represented, where the sign of the effect is determined by the model. We used a Bayesian inference model to determine the strengths and signs of causal effects. The statistical model outputs a posterior distribution for the regression coefficient, $\beta$, associated with each variable. For example, a model that includes both $\widehat{P}_{gadX}$ promoter activity and growth rate will have two coefficients, $\beta_{gadX}$ and $\beta_{growth}$, which indicate the impact of these two variables on survival. Predictions of positive values of $\beta$ are associated with variables where increased levels correspond to increased probability of survival following antibiotic exposure; negative predictions are associated with increased probability of death.

The survival outcome can be, in principle, determined by some, and not necessarily all, of the four possible variables: $\widehat{P}_{gadX}$, $\widehat{P}_{recA}$, and $\widehat{P}_{const}$ promoter activities, and growth rate. In addition, other unobserved and unmodeled factors, such as other cellular pathways, can also play a role in survival. To avoid overinterpreting a potentially misleading model, we considered all possible survival scenarios, with all possible combinations of variables (Fig. 4A). First, we considered four different models where only

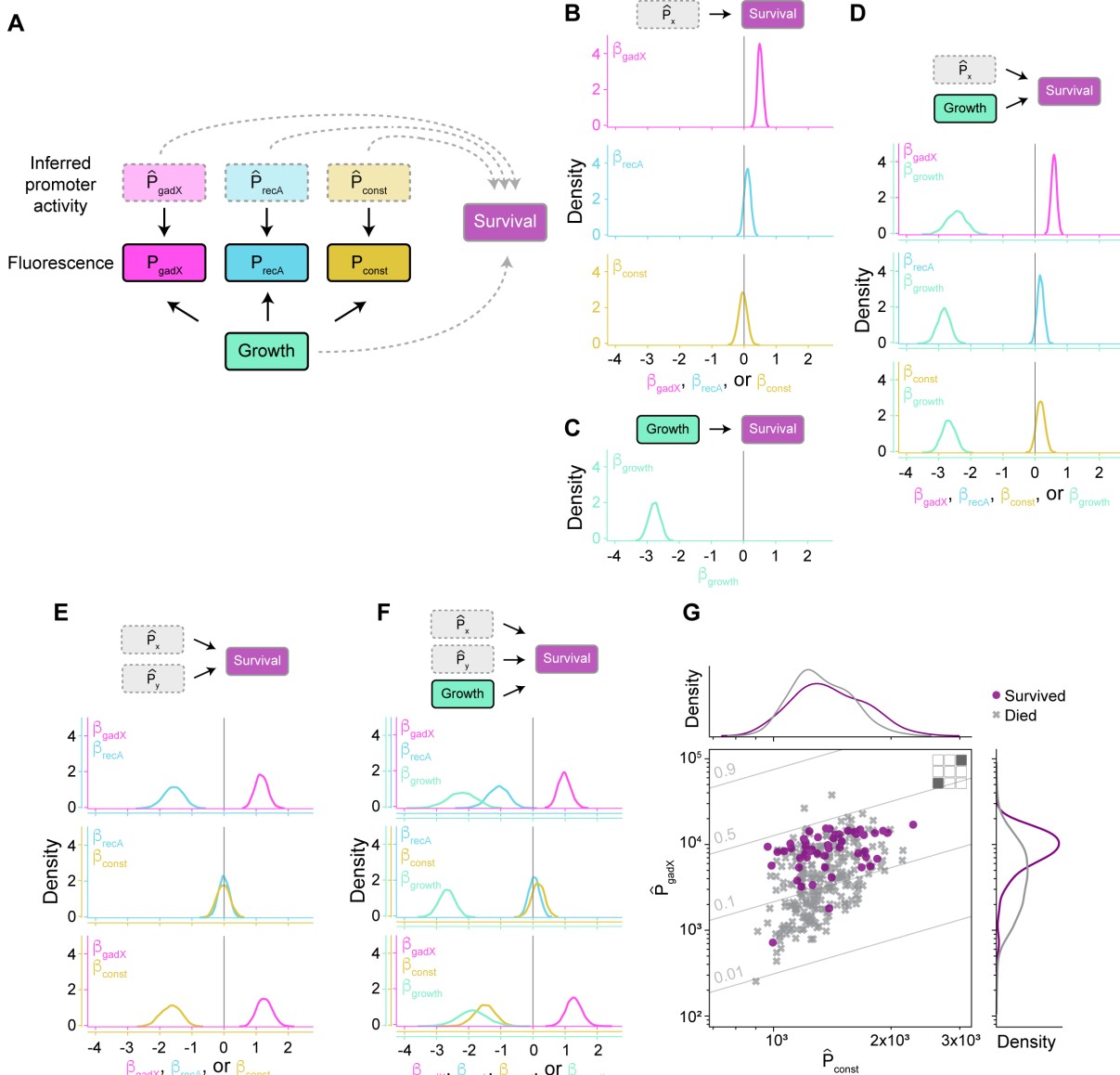

**FIG 4** Estimation of the causal effects of promoter activity and growth rate on survival. (A) Directed acyclic graph of the statistical model. Variables are represented in rectangles. The solid rectangles represent observed variables, while the dotted rectangles represent unobserved variables, which are inferred. The arrows between variables represent direct causal effects. The solid arrows represent effects included in all models, while the dotted arrows are included in only some of the models. (B) The posterior distributions for the regression coefficient $\beta_{gadX}$, $\beta_{recA}$, $\beta_{const}$, or (C) $\beta_{growth}$, when the regression contains only this one variable. The vertical line indicates zero effect. (D, E) Posterior distributions for regression coefficients when two variables (D, one promoter activity plus growth rate, and E, two promoter activities) are included in the regression. (F) Posterior distributions for regression coefficients when three variables are included in the regression. Note that we cannot show results for when all three promoter activities are included because cells contain at most two fluorescent reporters. (G) The distribution of inferred promoter activities for $\widehat{P}_{gadX}$ and $\widehat{P}_{const}$. The marginal distributions for cells that survived and died appear in the top and right panels as histograms. The thin gray diagonal lines indicate the probability of survival, computed by the logistic model with the average values of the posteriors (panel E, bottom plot). This panel includes data from $\widehat{P}_{gadX}$-CFP/$\widehat{P}_{const}$-YFP and $\widehat{P}_{const}$-CFP/$\widehat{P}_{gadX}$-YFP. Numerical values for the regression coefficients and their 94% credible intervals are listed in Table S4.

one variable has a direct causal effect on survival, considering the contributions of either the $\widehat{P}_{gadX}$, $\widehat{P}_{recA}$, or $\widehat{P}_{const}$ promoter activity, or growth rate in isolation (Fig. 4B and C). We used all the data associated with each promoter for this analysis. For example, for $\widehat{P}_{gadX}$, we used promoter activity data from the five dual reporter plasmids containing either $\widehat{P}$ gadX-CFP or $\widehat{P}_{gadX}$-YFP. For the growth rate, we used data from all nine strains. We found

that in the scenario where the model has only one variable with a direct causal effect on survival, the $\hat{P}_{gadX}$ pathway has a positive effect, with predictions for $\beta_{gadX} > 0$ (Fig. 4B). These results indicate that cells with higher $\hat{P}_{gadX}$ promoter activity are more likely to survive ciprofloxacin exposure. These results are consistent with prior findings on the relationship between $\hat{P}_{gadX}$ and antibiotic survival (16). By contrast, taken alone, $\hat{P}_{recA}$ and $\hat{P}_{const}$ have no measurable effect on survival, with $\beta_{recA}$ and $\beta_{const} \approx 0$. We also found that the growth rate has a strong negative effect on survival with $\beta_{growth} < 0$ (Fig. 4C), indicating that slow-growing cells are more likely to survive ciprofloxacin exposure. Thus, causal inference using models including only one variable shows a clear positive role for $\hat{P}_{gadX}$ and a negative role for growth rate in determining ciprofloxacin survival.

Next, we asked whether looking at combinations of variables would improve our ability to predict survival. In other words, if we considered models that allowed two or three variables, could we identify relationships where combinations of data improved survival predictions? We first considered the case where we used data for one promoter and growth rate, finding no major changes in the $\beta$ values associated with each variable (Fig. 4D). However, when we consider the case with two promoter activities simultaneously (with or without growth rate), we observed notable changes in the $\beta$ coefficients (Fig. 4E and F). Overall trends with $\beta_{gadX} > 0$ and $\beta_{growth} < 0$ were maintained, but surprisingly, for previously neutral $\hat{P}_{recA}$ and $\hat{P}_{const}$, we found that the results changed drastically when the regression considered multiple variables simultaneously. For example, when considering $\hat{P}_{gadX}$ paired with $\hat{P}_{const}$, the influence of $\hat{P}_{gadX}$ became slightly more pronounced than in the one-variable case, while the influence of $\hat{P}_{const}$ became strongly negative (Fig. 4E).

This discrepancy between scenarios where variables are analyzed alone or together can be understood if we consider the total and marginal distributions of cells that survived and died on these two variables. To illustrate this case, we focused on the relationship between $\hat{P}_{gadX}$ and $\hat{P}_{const}$ as a representative example (Fig. 4G). When considering only one variable ($\hat{P}_{gadX}$ or $\hat{P}_{const}$), the distributions of surviving and dead cells are clearly distinguishable for the $\hat{P}_{gadX}$ case (Fig. 4G, right density plots), while for $\hat{P}_{const}$, the histograms overlap nearly completely (Fig. 4G, top density plots). This view is responsible for the results of the regression in the case where a single variable is considered, where $\beta_{gadX} > 0$ and $\beta_{const} \approx 0$ (Fig. 4B). However, when extending to the two-variable case with $\hat{P}_{gadX}$ and $\hat{P}_{const}$, survival increases by progressing on both axes, which is illustrated by the diagonal lines indicating the probability of survival computed with the logistic regression model (Fig. 4G). This creates non-zero regression coefficients such that the model predicts $\beta_{gadX} > 0$ and $\beta_{const} < 0$ (Fig. 4E). Even though it might be tempting to think of the model with more variables as more accurate, it is not necessarily the case, as one of the variables could be a collider, opening a non-causal path in the graph. For example, in this two-variable model, an unobserved confounder could potentially be responsible for increased cell survival as well as increasing $\hat{P}_{gadX}$. In this case, the introduction of $\hat{P}_{gadX}$ in the regression would incorrectly attribute the survival effect of the confounder to $\hat{P}_{const}$ (assuming the existence of a direct or indirect causal link between $\hat{P}_{const}$ and $\hat{P}_{gadX}$). We also considered the case with three variables in the regression (Fig. 4F), finding that the addition of more variables does not resolve the issue, and the estimation of the effects remains sensitive to the variables included.

Overall, our results show consistent trends where $\hat{P}_{gadX}$ promoter activity is positively correlated with survival, and growth is negatively correlated. We assessed the predictive performance of these models with cross-validation (Fig. S10' supplemental text). This assessment shows good agreement with the conclusion that $\hat{p}_{gadX}$ promoter activity and growth, together, but especially combined, are good predictors of cell survival, unlike $\hat{P}_{recA}$ and $\hat{P}_{const}$ promoter activities. Counterintuitively, we find that adding more data, such as considering the data from two or more reporters of genes of interest, can

suggest spurious relationships and lead to inferences of causal connections that do not necessarily exist. These results serve as an important reminder that studies measuring subsets of regulatory networks need to be coupled with direct mechanistic understanding.

## DISCUSSION

Heterogeneity in the expression of stress response genes can allow a subset of cells within a population to exhibit antibiotic tolerance. To visualize these cell-to-cell differences in antibiotic survival and the expression of stress response genes that precede it, we used fluorescent reporters to follow single cells over time. We focused our analysis on dual color constructs that simultaneously report expression of two stress response pathways ($P_{gadX}$, $P_{recA}$) and a constitutive control ($P_{const}$). We found that there were clear differences in $P_{gadX}$ expression in the period prior to antibiotic exposure that were associated with whether cells survived or died following antibiotic treatment. We then focused on the period prior to antibiotic addition, using cross-correlation and autocorrelation analysis to identify temporal relationships between $P_{gadX}$, $P_{recA}$, $P_{const}$, and growth rate. We found clear anticorrelations between reporter levels and growth rate, consistent with a decrease in growth rate leading to an accumulation of cellular products. However, growth rate-dependent dilution effects are not likely to be the sole source of this delayed anticorrelation, as the three reporters exhibited different degrees of anticorrelation, with $P_{gadX}$ showing the strongest growth dependence. To normalize for growth rate effects and other confounding experimental factors, we next used a mathematical framework to infer promoter activity values derived from the experimental data. Using these inferred promoter activities, we asked how increased data from multiple stress response genes within single cells can increase our understanding of these phenotypic states related to antibiotic tolerance. Using a causal model, we found that the predicted relationship between promoter activity and survival is highly dependent on which data are provided to the model, and that spurious relationships can emerge as more data are added. Nevertheless, we did find consistent trends. For example, we found that $\widehat{P}_{gadX}$ correlates with increased ciprofloxacin survival, but the model suggests that the relationship between $\widehat{P}_{recA}$ and survival may not be as straightforward. Our results build on previous studies that use single-cell reporters to examine $\widehat{P}_{gadX}$ or $\widehat{P}_{recA}$ relationships with antibiotic survival and suggest that information about the mechanisms underlying these stress responses is necessary to draw conclusions about relationships with survival. Our data on the effect of growth rate on survival are consistent with previous studies suggesting that slow-growing cells allow for increased antibiotic tolerance (3, 22).

Our model also provides a useful method for normalizing fluorescence data from experiments with different imaging settings and removing the impact of growth to identify promoter activity. This framework for transforming data is especially useful in situations where reporter expression levels differ significantly, so the use of different microscopy imaging settings is necessary to capture the proper dynamic range of the reporters. These normalizations, along with growth rate normalizations, allowed us to combine data from all dual reporter plasmids into directly comparable promoter activity values.

There are several future directions for this work. For example, it would be interesting to test whether our results generalize to different antibiotics and different promoters to reveal additional phenotypic patterns that underlie antibiotic tolerance. In addition, our results identifying regression coefficients provide clear examples where additional data lead to incorrect inferences. It is possible to expand upon our initial model to explicitly model colliders or other regulatory network features.

Overall, our data and model demonstrate that capturing more information on gene expression states without the knowledge of possible confounding variables may not provide more accurate predictions about the relationship between gene expression and antibiotic survival. Fluorescence reporters are useful in capturing the dynamics

of genes of interest in single cells over time and can be multiplexed with the use of distinguishable fluorophores. The interpretation of data from multiple fluorescence reporters requires careful consideration to prevent researchers from drawing inaccurate conclusions about the relationship between gene expression and phenotypic outcomes like antibiotic survival.

## MATERIALS AND METHODS

### Plasmids and strains

Dual reporter plasmids were constructed starting from the *E. coli* promoter collection library by Zaslaver et al. 2006 (40). Plasmids from this collection contain the $P_{gadX}$ and $P_{recA}$ promoters driving the expression of the *gfpmut2* gene. For the constitutive reporter, a promoter engineered via point mutations from the T7 promoter (37) was introduced into the promoter-less vector from the Zaslavar collection (pUA66). The sequence for the constitutive promoter, which we refer to in this manuscript as $P_{const}$, is TTATCAAAAAGAGTA<u>TTGTCT</u>TAAAGTCTAACCTATAG<u>GAAAAT</u>TACAGCC**A**TCGAGAGGGAC ACGGCGAA, where the −35 and −10 sites are underlined and the +1 transcriptional start site is shown in bold.

The *gfpmut2* gene in the plasmids with $P_{gadX}$, $P_{recA}$, and $P_{const}$ was first replaced with *cfp* or *yfp* using Gibson assembly (41). We used this to make $P_X$-CFP and $P_Y$-YFP single reporter intermediates, where $P_X$ and $P_Y$ refer to $P_{gadX}$, $P_{recA}$, and $P_{const}$. Next, we constructed dual reporter plasmids by adding a $P_Y$-YFP cassette from one of these plasmids to a plasmid with a $P_X$-CFP cassette via Gibson assembly (41). Final plasmid constructs matching the grid in Fig. 1A are listed in Table S1. These dual reporter plasmids have the same low-copy origin (pSC101) and resistance gene (kanamycin cassette) as the original Zaslaver collection (40). The dual reporter plasmids were transformed into MG1655 *E. coli*.

Plasmids developed in this study are available from AddGene.

### Mother machine microfluidic experimental setup

Single colonies of MG1655 cells containing a dual reporter plasmid were grown overnight in 5 mL LB medium with 30 µg/mL kanamycin for plasmid maintenance. Overnight cultures were diluted 1/50 into 5 mL M9 medium. All M9 medium used contained M9 salts (0.1 mM CaCl$_2$, 2 mM MgSO$_4$, 0.4% glucose, and 0.2% casamino acids) with 30 µg/mL kanamycin and 0.5 g/L F-127 pluronic supplemented. Pluronic is used in microfluidic experiments to prevent cells from adhering to the microfluidic device outside of the desired chambers. This diluted culture was grown in a shaking incubator at 37°C for about 3 hours to reach an optical density at 600 nm (OD) of 1.0.

Once the cultures reached an OD of 1.0, cells were concentrated by spinning 1.5 mL of culture at 13,000 × *g* for 30 seconds. All but about 150–200 µL of the supernatant was decanted, and the pellet was resuspended in the remaining supernatant. The concentrated cultures were loaded into channels of a mother machine microfluidic device (30) using a 1 mL syringe. The device was then centrifuged at 4,000 × *g* for 5 minutes to load cells into the chambers. Once cells were loaded in the chambers, the device was placed on a Nikon Ti-E inverted fluorescence microscope. Fresh M9 medium with kanamycin and pluronic was provided to the cells via a peristaltic pump at a flow rate of 20 µL per minute.

### Time-lapse fluorescence microscopy

The mother machine devices were imaged using a 100× oil objective on the Nikon Ti-E inverted fluorescence microscope equipped with a perfect focus system (PFS). The microscope was enclosed in an incubated chamber set at 37°C for the duration of the experiment. For the first ~2 hours of growth in the mother machine, cells were grown on the microscope stage with medium flowing, which allowed the cells to recover from the

loading process without imaging. After recovery, images were acquired every 5 minutes using phase contrast and epifluorescence illumination in CFP, YFP, and RFP channels. Imaging settings were adjusted to keep fluorescence values within the detection range of the camera as the reporters have differences in brightness. The excitation light intensities (percentages) for the Lumencore light source and the exposure times (ms) for the ANDOR camera acquisition used are listed in Table S2. After imaging overnight to establish gene expression and growth rate histories for the cells, the cells were exposed to medium supplemented with 1 µg/mL ciprofloxacin and 1 µg/mL sulforhodamine 101 dye. This dye allowed us to track the presence of ciprofloxacin via red fluorescence (RFP imaging channel). After 10 minutes of exposure to medium with ciprofloxacin, the medium was switched back to the original medium without ciprofloxacin or sulforhodamine. Cells were maintained in the device for 18 hours and then assessed for survival or death in post-processing. Data for each reporter strain are derived from at least three biological replicates and include over 100 cells (Table S3).

## Image processing and data analysis

Data from the experiment were saved as an ND2 Nikon file. Images and videos were generated from these files using the Bioformats package on Fiji (ImageJ) software. We used the frames with red fluorescence in the media channel to determine the time of antibiotic addition, finding that it occurred over a period of three frames (10 minutes total). Kymographs were generated from ND2 files using MATLAB with the Bioformats package to capture single mother machine chamber images over time separately for phase contrast, CFP, and YFP.

For quantification of the data, the ND2 file was processed using the DeLTA software to segment and track cells and lineages while extracting data about growth rates, fluorescence, and division events (30, 31). Growth rate is defined at the single-cell level by the exponential rate of increase in the area of the cell within the microscopy image. If a cell has an area $A_0$ at time $t_0$ and an area $A_1$ at time $t_1$, then its growth rate between these two instants is computed as the logarithmic derivative of A with respect to $t$, or $\frac{\log A_1 - \log A_0}{t_1 - t_0}$. Given the predictable morphology of rod-like bacteria, computing growth rates based on the flat 2D projection of cells on microscopy images is common practice. Although we have presented the case with two data points here, growth rate calculations in DeLTA use additional refinements: the calculations use a centered finite difference scheme, and cell division events are handled as special cases that account for the length of cells before and after division. All data reported is from the mother cells within the chambers. After processing, we manually determined whether each of the mother cells resumed division after antibiotic removal (classified as a surviving cell), never resumed division after antibiotic removal (classified as a dead cell), or if the cells elongated and were swept out of the chamber (not included in the analysis because it was not possible to determine whether the cell ever resumed division).

To average data from multiple experiments onto the same plots, all data were analyzed beginning 10 hours prior to antibiotic addition and through the time of antibiotic addition. For time-lapse CFP and YFP plots, the fluorescence levels are raw fluorescence with background subtraction as explained in the next section. For growth rate plots, the growth rate from DeLTA was converted from "per frame" to "per hour" by multiplying by 12 (frames were every 5 minutes, $5 \times 12 = 60$ minutes).

For cross- and autocorrelations, the 10 hours of data prior to antibiotic addition were used. Growth rate (per hour) was first smoothed using a 12 frame window (1 hour) using the MATLAB smooth function. The cross-correlations between fluorescence and growth rate were computed using the MATLAB xcov function with the normalized method: xcov(fluorescence, growth rate, "normalized"). The fluorescence cross-correlations were calculated the same way as either xcov(CFP, YFP, "normalized") or xcov(YFP, CFP, "normalized"), as indicated in the legend. Autocorrelations were also computed

using the MATLAB xcov function with the normalized method, with only CFP, YFP, or growth rate as the input data.

Full-width at half maximum calculations were performed by identifying the peak of the autocorrelation signal (which occurs at a delay of zero), then calculating the width of the autocorrelation function at half the peak height.

## Background subtraction and gating

To remove background and outliers, we plotted fluorescence distributions with respect to cell area (Fig. S11). We found that we could model the fluorescence distributions as normal, provided that we removed the background and considered them on a logarithmic scale. In our analysis, we removed the background fluorescence by subtracting a value obtained by sampling at a location within the microfluidic chamber that did not contain cells. We used thresholds for cell area and fluorescence values to gate for cells and remove outliers. We determined these thresholds by bracketing the largest cloud of points in the (area, fluorescence) scatter plot. Outliers were spot-checked manually and correspond to cell fragments, debris, or bubbles within the microfluidic device, which we omit from the analysis via gating.

## Statistical model

We implemented the statistical analysis as a Bayesian model using the Python library PyMC (42). We used a generative model approach, defining variables and parameters as compositions of probabilistic distributions. The final variables (raw fluorescence) are defined as "observed," which corresponds to the experimental data inputs. Defining the model in this way builds a likelihood function of the parameters (unobserved variables) given the data (observed variables). PyMC then samples the Bayesian posterior distribution defined by this likelihood and the priors on the parameters and returns 1,000 samples of this distribution. Each of these samples is a full set of parameters, given according to their posterior probability. For every model, the sampling is done four independent times (four chains), which we compared to see whether the inference was robust, before combining them for further processing.

The complete model is made of two blocks corresponding roughly to the steps described in Fig. 3 (activity model) and Fig. 4 (survival model).

The activity model is itself composed of three successive modules, corresponding to the three biological/experimental steps identified in the main text. In all that follows, "i" is an index that identifies a cell, "prom" identifies a promoter (gadX, recA, or const), and "fluo" identifies a fluorescent reporter (CFP or YFP).

Promoter activity: we used $\mu_{prom} \sim$ Normal(6, 1), $\sigma_{prom} \sim$ HalfNormal(1), and $prom_{i,prom} \sim$ LogNormal($\mu_{prom}$, $\sigma_{prom}$). This section attributes to every promoter prom in every cell i, a value $prom_{i,prom}$ for "promoter activity" that is defined as a value on a log-normal distribution of parameters $\mu_{prom}$ and $\sigma_{prom}$ (which are common to all cells and only depend on the promoter). The prior distributions for the variables $\mu_{prom}$ and $\sigma_{prom}$ were chosen with the fewest number of assumptions possible: the choice of a Normal (respectively HalfNormal) distribution for a parameter taking real values (respectively, positive values including 0) is standard. The numerical values parameterizing these distributions were chosen with prior-predictive checks (43).

Reporter activity: we used $k_{prom,fluo} \sim$ ZeroSumNormal(1), $k'_{prom,fluo} \sim$ Normal(0, 1), $prot_{i,cfp} = prom_{i,cfp\_prom[i]} \cdot \exp(k_{cfp\_prom[i],cfp}) \cdot \exp(k'_{cfp\_prom[i],cfp} \cdot g_i)$, $prot_{i,yfp} = prom_{i,yfp\_prom[i]} \cdot \exp(k_{yfp\_prom[i],yfp}) \cdot \exp(k'_{yfp\_prom[i],yfp} \cdot g_i)$. In this section, the bookkeeping variables cfp_prom[i] and yfp_prom[i] are experimentally defined and indicate which promoter drives the expression of CFP (and YFP) in cell i. Two values of protein expression are defined for each cell, one for CFP ($prot_{i,cfp}$) and one for YFP ($prot_{i,yfp}$). They consist of the product of the relevant promoter activity, with adjustment factors k and k' to adjust for protein expression and dilution effects. Here also, the prior distributions for the adjustment variables $k_{prom,fluo}$ and $k'_{prom,fluo}$ are chosen with the fewest number of assumptions possible. Note that both of these notations define 6 scalar

variables, for example, for k′: $k'_{gadX,cfp}$, $k'_{gadX,yfp}$, $k'_{recA,cfp}$, etc, are each defined with a centered and unitary normal prior (Normal(0, 1)). For k, the use of a ZeroSumNormal prior adds the additional constraint that for every promoter, $k_{prom,cfp} + k_{prom,yfp} = 0$, that is, $k_{prom,cfp} = -k_{prom,yfp}$. Without this constraint, the model would try to infer two independent values for the CFP adjustment and the YFP adjustment. Without direct measurements of the promoter activity, we can only infer the difference between these adjustments, hence this constraint. The situation is different with k′ because several cells with different growth rates $g_i$ can be used to resolve the indeterminacy.

Fluorescence measurement: we used $k''_{cfp,cfp\_setting}$ ~ ZeroSumNormal(1), $k''_{yfp,yfp\_setting}$ ~ ZeroSumNormal(1), $\sigma'_{fluo}$ ~ HalfNormal(1), $fluo_{i,cfp} = prot_{i,cfp} \cdot exp(k''_{cfp,cfp\_setting[i]})$, $fluo_{i,yfp} = prot_{i,yfp} \cdot exp(k''_{yfp,yfp\_setting[i]})$, $meas_{i,fluo}$ ~ Normal($fluo_{i,fluo}$, $fluo_{i,fluo} \cdot \sigma'_{fluo}$). This section transforms the reporter activity in each cell $prot_{i,fluo}$ first into "exact" fluorescence values $fluo_{i,fluo}$ by applying an adjustment for each imaging setting, and then into the measured values $meas_{i,fluo}$ marred by multiplicative measurement noise. Here too, the adjustments for the imaging settings are defined with centered unitary priors summing to 0, for the same reason as explained in the previous paragraph.

The second block of the model is a logistic regression (Fig. 4A). It is defined as $\alpha + \Sigma_x \beta_x x$ where x can take different values depending on the model: growth rate, or $log(\hat{P}_{prom})$. This expression (survival logits) is then used to define a survival probability used in a Bernoulli trial (biased coin flip) describing the fate (survived or died) of each cell. The priors for α and the β coefficients are all unit normal (Normal(0, 1)).

Other scientific programming libraries used directly in this work include numpy (44), scipy (45), matplotlib (46), pandas (47), seaborn (48), xarray (49).

## Evaluation of the promoter activity inference model with a simulated data set

To evaluate the capacity of the Bayesian inference model to correctly infer promoter activity, we replicated our entire data set with a simulation. We used a stochastic telegraph model (50–52), a classical mechanistic simulation of promoter activity, transcription, and translation, to which we added cell growth with a fluctuating growth rate and cell division with a sizer mechanism (see supplemental text for details). The simulation was done with the rebop (53) library, a Rust implementation of the Gillespie algorithm (54). This enabled us to compare the actual promoter activity values to the values inferred by the model.

## Assessment of the prediction accuracy of the survival models

To assess the prediction accuracy of the survival models presented in Fig. 4, we used two standard methods: Leave-one-out cross-validation (LOO-CV) and the widely applicable information criterion (WAIC) (55, 56). The two approaches are asymptotically equivalent and give identical results on our models. To compute LOO-CV, we used the pareto-smoothed importance sampling approximation (PSIS) (57). Both these methods aim to estimate a quantitative indication of the accuracy of the out-of-sample performance prediction in terms of Bayesian deviance. In contrast to the standard practice of summing the contributions of every observation to the Bayesian deviance, here we averaged them to allow us to compare models that were run on different data sets. Further details are provided in the supplemental text. Both LOO-CV (PSIS) and WAIC were computed with the arviz library (58).

### ACKNOWLEDGMENTS

We thank Caroline Blassick, Cristian Coriano-Ortiz, and the other members of the Dunlop Lab for helpful discussions.

This work was supported by National Institutes of Health grant R01AI102922 and National Science Foundation grant MCB-2143289.

## AUTHOR AFFILIATIONS

[1]Department of Biomedical Engineering, Boston University, Boston, Massachusetts, USA
[2]Biological Design Center, Boston University, Boston, Massachusetts, USA

## AUTHOR ORCIDs

Mary J. Dunlop  http://orcid.org/0000-0002-9261-8216

## FUNDING

| Funder | Grant(s) | Author(s) |
| --- | --- | --- |
| National Institutes of Health | R01AI102922 | Mary J. Dunlop |
| National Science Foundation | MCB-2143289 | Mary J. Dunlop |

## AUTHOR CONTRIBUTIONS

Razan N. Alnahhas, Conceptualization, Investigation, Methodology, Visualization, Writing – original draft | Virgile Andreani, Conceptualization, Investigation, Methodology, Software, Visualization, Writing – original draft | Mary J. Dunlop, Conceptualization, Funding acquisition, Visualization, Writing – original draft

## DATA AVAILABILITY

The code for the simulation, inference, and figures is available as an online git repository: https://gitlab.com/dunloplab/dual-reporter-inference.

## ADDITIONAL FILES

The following material is available online.

### Supplemental Material

**Supplemental Information (mSystems01588-24-S0001.pdf).** Supplemental figures, tables, movie captions, text, and references.
**Movie S1 (mSystems01588-24-S0002.avi).** Cells growing in the mother machine microfluidic device with the $P_{gadX}$-CFP/$P_{gadX}$-YFP reporter plasmid.
**Movie S2 (mSystems01588-24-S0003.avi).** Cells growing in the mother machine microfluidic device with the $P_{gadX}$-CFP/$P_{recA}$-YFP reporter plasmid.
**Movie S3 (mSystems01588-24-S0004.avi).** Cells growing in the mother machine microfluidic device with the $P_{gadX}$-CFP/$P_{const}$-YFP reporter plasmid.
**Movie S4 (mSystems01588-24-S0005.avi).** Cells growing in the mother machine microfluidic device with the $P_{recA}$-CFP/$P_{gadX}$-YFP reporter plasmid.
**Movie S5 (mSystems01588-24-S0006.avi).** Cells growing in the mother machine microfluidic device with the $P_{recA}$-CFP/$P_{recA}$-YFP reporter plasmid.
**Movie S6 (mSystems01588-24-S0007.avi).** Cells growing in the mother machine microfluidic device with the $P_{recA}$-CFP/$P_{const}$-YFP reporter plasmid.
**Movie S7 (mSystems01588-24-S0008.avi).** Cells growing in the mother machine microfluidic device with the $P_{const}$-CFP/$P_{gadX}$-YFP reporter plasmid.
**Movie S8 (mSystems01588-24-S0009.avi).** Cells growing in the mother machine microfluidic device with the $P_{const}$-CFP/$P_{recA}$-YFP reporter plasmid.
**Movie S9 (mSystems01588-24-S0010.avi).** Cells growing in the mother machine microfluidic device with the $P_{const}$-CFP/$P_{const}$-YFP reporter plasmid.

## Open Peer Review

**PEER REVIEW HISTORY (review-history.pdf).** An accounting of the reviewer comments and feedback.

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
