## [Reviewer comments · mSystems]

Evaluating the predictive power of combined gene expression dynamics from single cells on antibiotic survival

Razan Alnahhas, Virgile Andreani, and Mary Dunlop

Corresponding Author(s): Mary Dunlop, Boston University

Review Timeline:

Submission Date:	November 23, 2024
Editorial Decision:	February 2, 2025
Revision Received:	April 3, 2025
Accepted:	April 16, 2025

Editor: Babak Momeni

Reviewer(s): The reviewers have opted to remain anonymous.

Transaction Report:

DOI: <https://doi.org/10.1128/msystems.01588-24>

Re: mSystems01588-24 (Evaluating the predictive power of combined gene expression dynamics from single cells on antibiotic survival)

Dear Dr. Mary Dunlop:

Two reviewers have evaluated the manuscript. While they find the manuscript's data useful and relevant, there are concerns about the rigor of definitions, the clarity of presentation, and adequate verification of the approach. The consensus is that these issues need to be addressed before the manuscript is published. Please find the detailed reviewers' evaluations below. Please also include a 'Data Availability' section (see below) in the revised manuscript.

Revision Guidelines

Sincerely,
Babak Momeni
Editor
mSystems

Reviewer #1 (Comments for the Author):

In their manuscript entitled "Evaluating the predictive power of combined gene expression dynamics from single cells on

antibiotic survival", Alnahas et al. investigate how concomitant single-cell reporter data for two stress-related genes can predict cellular outcomes (survival or death) after a brief exposure to ciprofloxacin. The reporters used are the *gadX* and *recA* promoters, which respectively reflect acid stress and DNA damage (SOS response). Using phase-contrast and fluorescence microscopy of cells confined within a mother machine microfluidic device, they capture detailed single-cell data. The authors start by showing that increased *P_{gadX}* expression prior to antibiotic exposure predicts survival outcomes. They show an anticorrelation between *P_{gadX}* expression and growth, where growth changes precede fluorescence changes, indicating that increased growth results in lowered expression. Finally the authors investigated whether their expression data and growth rate could predict survival outcomes upon antibiotic exposure, employing causal Bayesian modeling (single and multi-variable). The results suggests that *P_{gadX}* promoter activity is positively correlated with survival, while growth is negatively correlated with survival. Interestingly, their multi variable model raises concerns about colliders that distort causal relationships. The manuscript is well-written, though some of the figures could be clearer. While the methodological aspects, such as the dual-reporter system and causal modeling, represent a significant contribution, the biological insights themselves are not particularly novel, as the roles of stress response genes and growth rates in antibiotic survival have been previously explored.

Comments

The authors define heteroresistance as a temporary, non-heritable increase in tolerance in a subset of cells. This definition appears to align more closely with persistence rather than heteroresistance. Classically, heteroresistance refers to a transient increase in MIC in a subpopulation of cells. Conversely, persistence is typically described as a temporary, non-heritable state in which a small subpopulation of cells survives an otherwise lethal antibiotic dose without any changes in MIC. Since the authors focus on survival and regrowth/recovery after ciprofloxacin treatment, their work seems more aligned with studying persistence rather than heteroresistance. I recommend that the authors clarify their use of these terms and adjust their framing accordingly to ensure alignment with established definitions in the field. Of note here is that the 10-minute antibiotic exposure used in this study could raise questions about whether the authors are then studying "true" persisters.

The manuscript uses the term growth rate, but the definition and experimental determination of this parameter require further clarification. Is the growth rate of the population of cells that gave rise to the focal cell whose fluorescence is being measured? Is it an average fluorescence measurement of all cells descended from one cell with a specific growth rate? Or is it the elongation/volume growth rate of a single cell? A more precise description of how growth rate was measured would help readers understand the role of this parameter in the analysis.

The manuscript provides qualitative descriptions of the causal effects inferred from the Bayesian model, however it does not report key statistical metrics such as credible intervals, or posterior probabilities for the coefficients. Without these metrics, it is challenging to evaluate the statistical significance or robustness of the reported effects and models.

I was not able to access the online git repository (404: Page not found): <https://gitlab.com/dunloplab/dual-reporter-inference>. Please verify the link or provide an alternative method for accessing the repository.

Minor comments

The manuscript could benefit from streamlining Figures 1 and 2 by consolidating reciprocal measurements into single plots to enhance clarity. Could the promoter activity measurement not be used for this?

The manuscript would benefit from specifying the sample size, as this is essential for evaluating the robustness and reliability of the data. Given that variability is a key consideration in single-cell studies, knowing how many cells were assessed per experiment would provide valuable context for interpreting the results.

Reviewer #2 (Comments for the Author):

The authors use mother machine experiments to analyze how growth and stress response influence heteroresistance in *E. coli*. They measure cell growth and the expression of the *gadX* and *recA* stress response genes, along with a constitutive control, using dual reporter plasmids. They then carefully characterize the temporal correlations between these factors and develop a causal model to predict antibiotic survival based on them.

The main finding-that heteroresistance is primarily driven by *gadX* expression and slow growth-replicates previous studies. However, I believe the careful analysis and methodological improvements in this work provide a valuable contribution to the literature and will be of interest to researchers in both the antibiotic resistance and single-cell biology fields.

Overall, the manuscript is well written, the methods are clearly explained, and the conclusions are well supported by the data. However, I do have some minor concerns and questions that I believe should be addressed.

Major Comments:

1. The authors use a novel Bayesian model-based approach to infer promoter activities. While the approach seems appropriate, I would like to see some benchmarking data validating its accuracy. There are several possible options:

i) The inferred promoter activity could be compared to estimates obtained using the previously developed and more widely used approach by Locke et al. 2011 (<https://doi.org/10.1126/science.1208144>), which infers promoter activities from measured changes in fluorescence intensity.

ii) The method could be applied to a synthetic dataset created using a minimal mechanistic model of gene expression dynamics to show that it can accurately infer ground-truth promoter activity dynamics.

iii) The method could be tested against experimental data where promoter activities are manipulated in a controlled manner (e.g., using inducible promoter reporters).

Any of these approaches would sufficiently demonstrate the validity of the authors' method, and I leave it to the authors to choose the most suitable one.

2. Some of the statements in the text could benefit from further quantitative analysis. Specifically:

- For the cross-correlation analysis between fluorescence and growth (Fig. 2C), it would be highly beneficial to quantitatively analyze for which reporters the observed correlations deviate significantly from zero.
- The statement that the *gadX* autocorrelation function is wider than the others (line 218) was not immediately obvious from the figure to me. I suggest to quantitatively compare these widths, e.g. using the full-width at half-maximum (FWHM).
- The authors do not comment on the overall accuracy of their model (Fig. 4) in predicting cell survival. Providing statistics on model accuracy and comparing different models in this regard would be very informative.

Minor Comments:

- Figure 1D: The phase contrast image appears to be of rather low quality. This may be a PDF conversion issue, but please double-check the resolution of the source file.
- Figure 1E: To facilitate comparison, it would be helpful to use a consistent y-axis across all panels.
- Figure 2A: Are these traces of individual cells? Please clarify in the caption.
- Figures 2C and 2D: It would be useful to briefly comment on the cause of the positive cross-correlations at negative delays. I suspect these are not significant, but quantifying this and pointing it out explicitly would be beneficial (see also the second major comment above).
- Figures 3D and 3E: The axis labels do not clearly indicate that these figures show inferred promoter activity rather than raw fluorescence values. I suggest relabeling the axes to make this distinction explicit.
- Figure 3E (Optional Suggestion): This panel presents a large amount of data, making it somewhat difficult to extract overall patterns. I suggest to instead collapse the data by combining the fluorescent label swaps and instead show the data as in panel D, for the 3 possible combinations of reporters (*gadX-recA-growth*, *gadX-const-growth*, *recA-const-growth*). The current panel could then be added to SI instead.

Point-by-Point Response to Reviewers

Manuscript Number: mSystems01588-24

We appreciate the reviewers' comments and suggestions for improving the content and clarity of the manuscript. We have addressed their concerns in the revised manuscript. A complete list of changes and detailed responses to the comments follows.

Reviewer #1

In their manuscript entitled "Evaluating the predictive power of combined gene expression dynamics from single cells on antibiotic survival", Alnahas et al. investigate how concomitant single-cell reporter data for two stress-related genes can predict cellular outcomes (survival or death) after a brief exposure to ciprofloxacin. The reporters used are the *gadX* and *recA* promoters, which respectively reflect acid stress and DNA damage (SOS response). Using phase-contrast and fluorescence microscopy of cells confined within a mother machine microfluidic device, they capture detailed single-cell data. The authors start by showing that increased *P_{gadX}* expression prior to antibiotic exposure predicts survival outcomes. They show an anticorrelation between *P_{gadX}* expression and growth, where growth changes precede fluorescence changes, indicating that increased growth results in lowered expression. Finally the authors investigated whether their expression data and growth rate could predict survival outcomes upon antibiotic exposure, employing causal Bayesian modeling (single and multi-variable). The results suggests that *P_{gadX}* promoter activity is positively correlated with survival, while growth is negatively correlated with survival. Interestingly, their multi variable model raises concerns about colliders that distort causal relationships.

The manuscript is well-written, though some of the figures could be clearer. While the methodological aspects, such as the dual-reporter system and causal modeling, represent a significant contribution, the biological insights themselves are not particularly novel, as the roles of stress response genes and growth rates in antibiotic survival have been previously explored.

Comments

The authors define heteroresistance as a temporary, non-heritable increase in tolerance in a subset of cells. This definition appears to align more closely with persistence rather than heteroresistance. Classically, heteroresistance refers to a transient increase in MIC in a subpopulation of cells. Conversely, persistence is typically described as a temporary, non-heritable state in which a small subpopulation of cells survives an otherwise lethal antibiotic dose without any changes in MIC. Since the authors focus on survival and regrowth/recovery after ciprofloxacin treatment, their work seems more aligned with studying persistence rather than heteroresistance. I recommend that the authors clarify their use of these terms and adjust their framing accordingly to ensure alignment with established definitions in the field. Of note here is that the 10-minute antibiotic exposure used in this study could raise questions about whether the authors are then studying "true" persisters.

We appreciate this comment and wish to avoid possible confusion with the terms we use. We also agree that our short-duration antibiotic exposure is not consistent with true persisters. We have edited the manuscript so that we now include discussion of both heteroresistance and persistence in the introduction, but we use the more general term of

tolerance for what we present, and we are clear about the type of transient tolerance we focus on in our experiments.

→ See Abstract, Introduction, and various places throughout the manuscript

The manuscript uses the term growth rate, but the definition and experimental determination of this parameter require further clarification. Is the growth rate of the population of cells that gave rise to the focal cell whose fluorescence is being measured? Is it an average fluorescence measurement of all cells descended from one cell with a specific growth rate? Or is it the elongation/volume growth rate of a single cell? A more precise description of how growth rate was measured would help readers understand the role of this parameter in the analysis.

Thank you for catching this omission. We now include a precise definition of growth rate in the Methods.

→ See Methods “Image processing and data analysis”

The manuscript provides qualitative descriptions of the causal effects inferred from the Bayesian model, however it does not report key statistical metrics such as credible intervals, or posterior probabilities for the coefficients. Without these metrics, it is challenging to evaluate the statistical significance or robustness of the reported effects and models.

These data are presented graphically in Fig. 4, where the plots present the posterior distributions for the regression coefficients. We have added a supplementary table that lists regression coefficients and credible intervals for the models.

→ See Fig. 4

→ See Table S4

I was not able to access the online git repository (404: Page not found): <https://gitlab.com/dunloplab/dual-reporter-inference>. Please verify the link or provide an alternative method for accessing the repository.

Thank you for pointing out this oversight. We fixed the access to make the repository public.

→ See Code Availability

Minor comments

The manuscript could benefit from streamlining Figures 1 and 2 by consolidating reciprocal measurements into single plots to enhance clarity. Could the promoter activity measurement not be used for this?

We appreciate the concern about the amount of data presented in Figures 1 and 2, however, in Fig. 1 we felt it was important to present the original fluorescence data of all the reporter combinations, including the reciprocals, in order to show that the fluorophore

arrangements do not affect the overall expression patterns from each promoter or their correlation with cell survival. In Fig. 2, we use raw fluorescence data for the cross correlations as the promoter activity is already normalized to growth rate, so it would not be appropriate to use this for growth rate cross correlation calculations.

The manuscript would benefit from specifying the sample size, as this is essential for evaluating the robustness and reliability of the data. Given that variability is a key consideration in single-cell studies, knowing how many cells were assessed per experiment would provide valuable context for interpreting the results.

We have added this information in the Methods and Table S3.

- See Methods “Time-lapse fluorescence microscopy”
- See Table S3

Reviewer #2

The authors use mother machine experiments to analyze how growth and stress response influence heteroresistance in *E. coli*. They measure cell growth and the expression of the *gadX* and *recA* stress response genes, along with a constitutive control, using dual reporter plasmids. They then carefully characterize the temporal correlations between these factors and develop a causal model to predict antibiotic survival based on them.

The main finding—that heteroresistance is primarily driven by *gadX* expression and slow growth—replicates previous studies. However, I believe the careful analysis and methodological improvements in this work provide a valuable contribution to the literature and will be of interest to researchers in both the antibiotic resistance and single-cell biology fields.

Overall, the manuscript is well written, the methods are clearly explained, and the conclusions are well supported by the data. However, I do have some minor concerns and questions that I believe should be addressed.

Major Comments:

1. The authors use a novel Bayesian model-based approach to infer promoter activities. While the approach seems appropriate, I would like to see some benchmarking data validating its accuracy. There are several possible options:
 - i) The inferred promoter activity could be compared to estimates obtained using the previously developed and more widely used approach by Locke et al. 2011 (<https://doi.org/10.1126/science.1208144>), which infers promoter activities from measured changes in fluorescence intensity.
 - ii) The method could be applied to a synthetic dataset created using a minimal mechanistic model of gene expression dynamics to show that it can accurately infer ground-truth promoter activity dynamics.
 - iii) The method could be tested against experimental data where promoter activities are manipulated in a controlled manner (e.g., using inducible promoter reporters).

Any of these approaches would sufficiently demonstrate the validity of the authors' method, and I leave it to the authors to choose the most suitable one.

We reproduced our entire dataset in silico, with parameters chosen to match the experimental dataset as closely as possible (following approach “ii”) suggested by the reviewer’s comment). We analyzed this simulated dataset exactly as we analyzed the experimental one and found that the inferred promoter activities match the actual promoter activities well. We have added two supplementary figures showing how the simulation parameters relate to the observed distributions (Fig. S7), and the match between the inferred promoter activity and the actual (simulated) recent promoter activity (Fig. S8). In addition, we describe the model and parameters in Methods and Supplementary Text.

→ See Fig. S7

→ See Fig. S8

→ See Methods “Evaluation of the promoter activity inference model with a simulated dataset”

→ See Supplementary Text “Assessing the accuracy of the promoter activity inference model”

2. Some of the statements in the text could benefit from further quantitative analysis. Specifically:

- For the cross-correlation analysis between fluorescence and growth (Fig. 2C), it would be highly beneficial to quantitatively analyze for which reporters the observed correlations deviate significantly from zero.

We have added statistical significance tests to our analysis of the data in Fig. 2C. We identified the extrema of the function (the minimum for the cross correlations between fluorescence and growth) and checked for statistical deviations from zero. The results of this analysis are now included in Fig. 2C. We found that many of the cross correlation curves did not have statistically significant deviations from zero. This is likely due to the fact that our single-cell resolution measurements are noisy and technically challenging experiments make it difficult to obtain data in high throughput. However, the fact that we observed broadly consistent trends in our results gives us confidence in these data. Nevertheless, we have updated the Results to use descriptive language that does not imply statistical significance.

→ See Fig. 2C

→ See Results “Temporal relationships between expression and growth”

- The statement that the gadX autocorrelation function is wider than the others (line 218) was not immediately obvious from the figure to me. I suggest to quantitatively compare these widths, e.g. using the full-width at half-maximum (FWHM).

Thank you for this suggestion. We calculated the FWHM for the correlation functions for all reporters to provide a quantitative comparison of the autocorrelations. We now include this as a new figure panel in Fig. 2F.

→ See Fig. 2F

→ See Results “Temporal relationships between expression and growth”

→ See Methods “Image processing and data analysis”

- The authors do not comment on the overall accuracy of their model (Fig. 4) in predicting cell survival. Providing statistics on model accuracy and comparing different models in this regard would be very informative.

To address this point we compared the prediction accuracy of the survival models with the standard techniques of Leave-one-out cross-validation (LOO-CV), and the Widely Applicable Information Criterion (WAIC). We additionally compared the performance of our models to the performance of a random maximum entropy model. All methods show that P_{gadX} and growth, together and combined, are good predictors of cell survival, unlike P_{recA} and P_{const} . We thank the reviewer for this suggestion, as we believe that this analysis represents a meaningful addition to our article.

→ Fig. S8

→ See Methods “Assessment of the prediction accuracy of the survival models”

→ Supplementary Text “Assessing the predictive accuracy of the survival models”

Minor Comments:

- Figure 1D: The phase contrast image appears to be of rather low quality. This may be a PDF conversion issue, but please double-check the resolution of the source file.

Thank you for bringing this to our attention. We have updated the figure.

→ Fig. 1D

- Figure 1E: To facilitate comparison, it would be helpful to use a consistent y-axis across all panels.

We have fixed the axis scales.

→ See Fig. 1E

- Figure 2A: Are these traces of individual cells? Please clarify in the caption.

Yes, these are single cell traces. We have clarified this in the caption and the text.

→ See Fig. 2A caption

→ See Results “Temporal relationships between expression and growth”

- Figures 2C and 2D: It would be useful to briefly comment on the cause of the positive cross-correlations at negative delays. I suspect these are not significant, but quantifying this and pointing it out explicitly would be beneficial (see also the second major comment above).

We now address this briefly in the text.

→ See Results “Temporal relationships between expression and growth”

- Figures 3D and 3E: The axis labels do not clearly indicate that these figures show inferred promoter activity rather than raw fluorescence values. I suggest relabeling the axes to make this distinction explicit.

Thank you for this suggestion. We have updated the notation throughout the manuscript such that raw fluorescence values are denoted with P (e.g. P_{gadX}) and the associated inferred promoter activities are denoted with \hat{P} (e.g. \hat{P}_{gadX}). This change has been implemented in Fig. 3 and 4 and throughout the text of the manuscript.

→ See Fig. 3

→ See Fig. 4

→ See Results

→ See Methods

- Figure 3E (Optional Suggestion): This panel presents a large amount of data, making it somewhat difficult to extract overall patterns. I suggest to instead collapse the data by combining the fluorescent label swaps and instead show the data as in panel D, for the 3 possible combinations of reporters (gadX-recA-growth, gadX-const-growth, recA-const-growth). The current panel could then be added to SI instead.

We carefully considered this suggestion, as we agree that Fig. 3E includes a large amount of data. Ultimately, alternative representations were less clear than the original representation, so we have elected to maintain the original figure format.

Re: mSystems01588-24R1 (**Evaluating the predictive power of combined gene expression dynamics from single cells on antibiotic survival**)

Dear Dr. Dunlop:

Your manuscript has been accepted, and I am forwarding it to the ASM production staff for publication. Your paper will first be checked to make sure all elements meet the technical requirements. ASM staff will contact you if anything needs to be revised before copyediting and production can begin. Otherwise, you will be notified when your proofs are ready to be viewed.

While the points raised by the reviewers have been adequately addressed, there is one remaining point that I would encourage you to address before publication. The added definition of growth rate based on the cell area in 2D images will likely be different by a factor from the growth rate definition based on the cell biomass (or cell number), because the third dimension is not accounted for. I suggest adding a note to your definition to clarify this difference.

Cover Image Submissions: If you would like to submit a potential Cover Image, please email a file and a short legend to mssystems@asmusa.org. Please note that we can only consider images that (i) the authors created or own and (ii) have not been previously published. By submitting, you agree that the image can be used under the same terms as the published article. Image File requirements: TIF/EPS, 7.5 inches wide by 8.25 inches tall (at least 2,250 pixels wide by 2,475 pixels tall), minimum 300 dpi resolution (600 dpi preferred), RGB, and no figure elements, e.g., arrows or panel labels. The legend should be a short description of the image, 1-2 sentences recommended. Please download and use this interactive template in Adobe to ensure that your proposed cover image meets our size requirements (<https://journals.asm.org/pb-assets/pdf-text-excel-files/ASM-Interactive-Sizing-Cover-Template-1715689791.pdf>).

We recognize that the video files can become quite large, so to avoid quality loss ASM suggests sending the video file via <https://www.wetransfer.com/>. When you have a final version of the video and the still ready to share, please send it to mSystems

staff at mSystems@asmusa.org.

Sincerely,
Babak Momeni
Editor
mSystems